# The Structure of Biologically Active Functionalized Azoles: NMR Spectroscopy and Quantum Chemistry

Lyudmila I. Larina

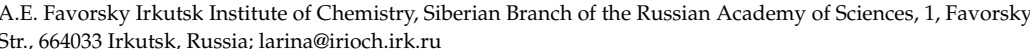

A.E. Favorsky Irkutsk Institute of Chemistry, Siberian Branch of the Russian Academy of Sciences, 1, Favorsky Str., 664033 Irkutsk, Russia; larina@irioch.irk.ru

**Abstract:** This review summarizes the data on the stereochemical structure of functionalized azoles (pyrazoles, imidazoles, triazoles, thiazoles, and benzazoles) and related compounds obtained by multipulse and multinuclear $^1$H, $^{13}$C, $^{15}$N NMR spectroscopy and quantum chemistry. The stereochemistry of functionalized azoles is a challenging topic of theoretical research, as the correct interpretation of their chemical behavior and biological activity depends on understanding the factors that determine the stereochemical features and relative stability of their tautomers. NMR spectroscopy, in combination with quantum chemical calculations, is the most convenient and reliable approach to the evaluation of the stereochemical behavior of, in particular, nitrogen-containing heteroaromatic and heterocyclic compounds. Over the last decade, $^{15}$N NMR spectroscopy has become almost an express method for the determination of the structure of nitrogen-containing heterocycles.

**Keywords:** functionalized azoles; pyrazoles; imidazoles; triazoles; thiazoles; benzazoles; stereochemical features; multinuclear and 2D NMR spectroscopy; quantum chemical calculations

## 1. Introduction

The derivatives of azoles and their annulated analogs, benzazoles, occupy an important place in heterocyclic chemistry, and have extensive applications in numerous fields of materials science, industry, agriculture, biology, and organic and medical chemistry [1–17]. Due to their intensive use in various branches of medicine, technology and agriculture, azoles attract the closest attention of researchers. Different monographs and reviews are devoted to the chemistry, properties and application of azoles [18–34].

Our long-term studies of functionalized azoles and relative compounds were published in a monograph on nitroazoles [35], as well as several reviews devoted to the chemistry of five-membered azoles and their condensed (annelated) analogs [36–39], organosilicon azoles [40], the NMR spectroscopy and mass-spectrometry of nitroazoles [41,42], and the structure and electronic effects of five-membered nitrogenated heterocycles [43]. In addition, we have summarized the results of studies on tautomerism and the structure of functionalized azoles [44–47]. The problems of prototropy of NH-unsubstituted azoles [45] and silylotropy of *N*-silylated analogs [47] were discussed and analyzed in detail using the methods of multinuclear and dynamic NMR spectroscopy. The results of the study of the tautomerism and structural features of functional azoles in the solid state by the Nuclear Quadrupole Resonance method [46] were also surveyed.

The unique properties of the azole cycle, such as the heteroaromatic nature of the ring, its high chemical stability, the ability to form hydrogen bonds, and its large dipole moment are effectively used in fine organic synthesis for the design of novel materials and biologically active molecules. Such widespread application of azoles requires an understanding of the peculiarities of their electron structure, spectral features, stereochemical behavior and reactivity.

The azoles have the middle position, because they do not possess evidently expressed π-donating or π-deficient properties. This classification reflects the π-electron density

distribution in the ground state of a molecule. Although the reactivity is determined by the energy difference between the ground and transition reaction state, in practice the correlation of π-excessive change and the facility of electrophilic substitution are frequently observed. Indeed, when the number of "pyridine" nitrogen atoms increases, the π-donating properties of azoles decrease, and thus their reactivity in electrophilic substitution reactions is reduced [6,45,48].

NMR spectroscopy, as already indicated [41,44,45,47], is one of the most convenient and effective methods for the study of the stereoelectronic structure, and the stereodynamic and chemical behavior of organic and, in particular, heterocyclic compounds. Due to the rapid development of NMR experimental techniques, studies on [13]C, [15]N, [17]O and [29]Si nuclei—necessary for functionally substituted azoles (and heterocycles in general)—have become available and even rooting. Nitrogen NMR spectroscopy allows one to obtain direct information on the state of the nitrogen atoms of both the heteroaromatic ring and nitrogen-containing substituents; thus, in combination with [1]H and [13]C NMR data, it is possible to most fully reveal the dependences connecting the spectral parameters with the structure and electronic effects in the series of azoles.

Structural and stereochemical studies of azoles lacking tautomeric transitions (or *N*-protected), the most important components of different drug (or drug precursors), have not been performed or summarized.

The present review is devoted to structural studies of functionalized azoles, mainly *N*-protected azoles or those containing no a pyrrole nitrogen atom (thiazoles, oxazoles, etc.) by multinuclear NMR spectroscopy and quantum chemistry. Azoles are extensively used as energy materials, medicines, radiosensitizers, ionic liquids, plasticizers, dyes, plant growth regulators, pesticides and herbicides, high universal bases in peptide nucleic acids, synthons for fine organic synthesis, and precursors for nano-composites [35,49–51].

The study of the structure and stereochemical behavior of *N*-substituted azoles is of particular interest, as the composition of drugs mainly contains non-tautomeric analogs [4,5,35,52,53]. It is known that the presence of tautomerism in azoles leads to certain difficulties in the design and creation of drugs based on them [52,54–56]. In the estimation of the influence of tautomerism on biological activity, it is necessary to consider both the thermodynamic and kinetic aspects of tautomeric equilibrium, and also to take into account the effect of the medium pH, the solvent properties (polarity), and temperature. It is very important to recognize the tautomerism potential in heterocyclic compounds, and to evaluate the role of individual tautomers in biological action. Almost always, the effect of a single tautomer on therapeutic activity is determined by the timescale of the tautomeric (prototropic) equilibrium relative to that of the biological process in question [52].

As such, stereochemical studies of azoles with an *N*-protected structure are of great interest. In these cases, additional difficulties caused by prototropy, as in the case of *N*-unsubstituted examples, are not superimposed. On the other hand, the absence of tautomerism in the molecule leads to a violation of the symmetry and a complication of the NMR spectra. Therefore, in order to establish the structure of the compounds, it is necessary to use the methods of multinuclear [1]H, [13]C, [15]N and [29]Si NMR spectroscopy, as well as unconventional methods of 1D INADEQUATE (Incredible Natural Abundance DoublE QUAntum Transfer Experiment) or 2D (multipulse) NMR spectroscopy (COSY, NOESY, HSQC, HMBC, ROESY).

## 2. The Products of the Vicarious Nucleophilic Substitution of Hydrogen in *N*-organyl-Substituted Nitroazoles

This section may be divided by subheadings. It should provide a concise and precise description of the experimental results, their interpretation, and the experimental conclusions that can be drawn.

The nitro derivatives of *N*-protected (substituted) azoles, which occupy a special place in the chemistry of heterocyclic compounds, are widely used in the reaction of nucleophilic aromatic substitution of hydrogen. They are key structural motifs for a huge number

of drugs: azomycin, carnidazole, dimetridazole, flunidazole, megazole, metronidazole, misonidazole, nitazole, ornidazole, ronidazole, tinidazole, etc. [14,16,35,49]. The products of the *C*-amination of *N*-substituted nitroazoles have been recognized as high-energy compounds [57,58].

In 1978, as a result of the search for new forms of interaction of electron-deficient aromatic systems with C-nucleophiles, leading to anionic σ-complexes, Makosha proposed an interesting type of reaction of aromatic nucleophilic substitution, which continues to be successfully developed [59–63]. In the course of the reaction, the X–anion is eliminated from the formed σH–adduct instead of the elimination of the hydride anion from the ring (Scheme 1). In this case, the X–anion is a vicarious ("substitute", "acting") leaving group. For this reason, the process was called the "vicarious nucleophilic substitution (VNS) of hydrogen" [44,59–67].

**Scheme 1.** VNS reaction on the example of nitroarenes. X, leaving group; Y, carbanion stabilizing group; R, a substituent; Z, substituent in the nitroaromatic ring.

A characteristic feature of VNS is that, in the case of nitroarenes containing leaving groups such as halogens, etc., as a rule, it proceeds much faster than the usual nucleophilic substitution of $S_N$Ar for the same groups. This is due to the fact that carbanions are much faster attached to carbon carrying a hydrogen atom rather than a substituent, as well as a relatively high rate of HX elimination from the σ$^H$-adduct under suitable conditions (an excess of the base that carries out this elimination).

The most widely studied are the reactions of nitroarenes and nitroheteroarenes with carbanions containing halogens, PhS, PhO, $R_2$NC(S)S, etc., as a leaving group X, and stabilizing groups Y-CN, COOR, $SO_2$Ar, $SO_2$OR, $SO_2NR_2$, and SOAr, P(O)(OEt)$_2$, etc. Almost any combination of such substituents gives a carbanion which is capable of entering into the VNS reaction, although in reality there are some limitations due to the following factors:

● the high activity of the carbanion precursor as alkylating agent;
● the instability of the carbanion or its low nucleophilicity, when the X, Y and R groups effectively delocalize the negative charge.

The substituents in nitroarenes of both electron-donor (OR, NR$_2$, SR, Alk) and electron-acceptor (COOR, CF$_3$, NO$_2$, etc.) characters do not interfere with the reaction, exerting only an orienting effect.

Thus, the VNS reaction is practically the only method for the direct introduction of an amino group or other functional groups (aldehyde, cyanomethyl, organylsulfonylmethyl, trihalomethyl, etc.) into aromatic and heterocyclic compounds. A typical example of such a process is the reaction of nitrobenzene with chloromethylphenylsulfone, which results in a mixture of *o*- and *p*-nitrobenzylphenylsulfones (Scheme 2) [59,64–80]. Therefore, halogen acts as a leaving group instead of the ring hydrogen, which is incapable of elimination as a hydride anion.

**Scheme 2.** Formation of *o*- and *p*-nitrobenzylphenylsulfones in the reaction of nitrobenzene with chloromethylphenylsulfone.

In reactions of this type, the substitution of a hydrogen atom with a carbanion residue occurs in the substrate, and the halide anion, rather than hydrogen, leaves the intermediate σ-adduct.

It should be noted that the process of the vicarious *C*-amination of *N*-protected nitroazoles, which are promising drugs, has not been insufficiently studied.

*2.1. Structure of the C-Amination Products of N-Substituted Nitroazoles with 1,1,1-trimethylhydrazinium Halides*

Unsymmetric dimethylhydrazine (1,1-dimethylhydrazine, *N,N*-dimethylhydrazine, UDMH) is a well-known product which is widely used as the component of liquid rocket fuel. Nevertheless, the possibilities of UDMH application for the preparation of various valuable products attracts ever-increasing interest.

Previously, within the framework of the International Project (ISTC 427) on the utilization of rocket fuel, we dealt with the deactivation of the toxic asymmetric dimethylhydrazine (heptyl) by treatment with alkyl halides. The method is based on the fact that alkyl halides react at a high rate, selectively and almost quantitatively with 1,1-dimethylhydrazone, to form non-toxic 1-alkyl-1,1-dimethylhydrazinium halides (Scheme 3).

R = Alkyl, Hal = Cl, Br, I;   R = CH$_3$, Hal = I

**Scheme 3.** The quaternization of dimethylhydrazine.

Trialkylhydrazine halides are widely used in organic synthesis as aminating agents in the VNS reaction in nitroaromatic and nitroheterocyclic compounds. The technique of VNS amination is a very convenient, and in some cases it is the only method for the direct introduction of an amino group into heteroaromatic systems.

The vicarious nucleophilic substitution of hydrogen, in particular *C*-amination, is typical mainly for *N*-substituted azoles, in which both prototropic rearrangements and the formation of azolyl cations are excluded. The products of the direct amination of 1-organyl-substituted nitroazoles and their model compounds (nitrobenzenes) with 1,1,1-trimethylhydrazinium halides or 4-amino-1,2,4-triazole in a superbasic medium using VNS methodology have been studied by multipulse and multinuclear $^1$H, $^{13}$C and $^{15}$N NMR, ESR spectroscopy, and quantum chemistry [44,64,78,81–92].

In particular, we studied the *C*-amination of nitroazoles with 1,1,1-trimethylhydrazinium halides using the VNS amination technique, namely, 1-methyl-4-nitropyrazole (**1**), 1-methyl-4-nitroimidazole (**2**), 2-phenyl-4-nitro-1,2,3-triazole (**3**), 1-methyl-5-nitrobenzimidazole (**4**) and 1-methyl-6-nitrobenzimidazole (**5**) (Scheme 4, Table 1).

**Scheme 4.** Vicarious nucleophilic *C*-amination of nitroazoles in a superbasic medium using 1,1,1-trimethylhydrazinium halides.

Analysis of the $^1$H, $^{13}$C and $^{15}$N NMR spectra of the initial nitro compounds and amination products showed that the amino group in the five-membered nitroazoles **1**, **2**, and **3** enters position 5, and in nitrobenzimidazoles (**4**, **5**) it enters either position 4 or 7 [35,44,82,83,86] (Table 1). As a result of the reaction, 5-amino-1-methyl-4-nitropyrazole (**6**), 5-amino-1-methyl-4-nitroimidazole (**7**), 2-phenyl-4-nitro-5-amino-1,2,3-triazole (**8**), 4-amino-1-methyl-5-nitrobenzimidazole (**9**), and 7-amino-1-methyl-6-nitrobenzimidazole (**10**) are obtained.

In the proton spectra of these compounds, the broad signals of the protons of the amino group are found in the range of 7.3 ÷ 7.7 ppm (Table 1). In the $^{15}$N NMR spectrum, upfield signals in the region $-300 ÷ -317$ ppm are characteristic of the nitrogen atom of the amino group [35,41,44,45,49,93]. Signals in the range of $-197 ÷ -227$ and $-70 ÷ -135$ ppm refer to the "pyrrole" (N-1) and "pyridine" (N-2) nitrogen atoms, respectively. The nitrogen nucleus of the nitro groups resonates in a much lower field ($∼-10 ÷ -30$ ppm). The 13C NMR signals of the carbon bound to the nitro group are usually broadened or do not appear at all due to the quadrupole broadening [35,41,93] (for this reason, the 13C signals of the carbon atoms of C-NO2 are absent in some spectra of the compounds) (Table 1). This broadening disappears when the 13C-{1H, 14N} triple resonance technique is used [94].

Furthermore, 1-Methyl-5-nitrobenzimidazole (**4**) and 1-methyl-6-nitrobenzimidazole (**5**) are aminated by 1,1,1-trimethylhydrazinium iodide (TMHI) in *t*-BuOK/DMSO (10 h) at positions 4 and 7, to form 1-methyl-4-amino-5-nitrobenzimidazole (**9**) and 1-methyl-4-amino-6-nitrobenzimidazole (**10**) (Scheme 4, Table 1). The structure of **9** is confirmed by the presence in the proton spectra of two doublets with $^3J$ = 6.8 Hz, which are related to H-6 and H-7. Similarly, two doublets ($^3J$ = 6.9 Hz) in the proton spectra of benzimidazole **10** are attributed to the H-4 and H-5 protons. The 2D NOESY spectra of compound **9** show cross peaks of the protons at position 7 with protons of the methyl group at position 1, while for **10** a cross peak of protons of the methyl group and the amino group is found. The introduction of an amino group into the phenylene fragment of nitrobenzimidazoles **4** and **5** only slightly increases the screening of the nitrogen-15 nuclei of the hetero ring (~6–7 ppm-N-1, N-3) (Table 1).

In addition, under the more harsh conditions of the VNS reaction, the amination of 1-methyl-6-nitrobenzimidazole is accompanied by the formation of two amination products at positions 7 (**10**) and 2 (**11**), in a ratio of 2:1, respectively (Scheme 5, Table 1). [44,90,91].

**Scheme 5.** VNS amination of 1-methyl-6-nitrobenzimidazole in a superbasic medium using 1,1,1-trimethylhydrazinium iodide with formations **10** and **11**.

The structure of these compounds was proved by 2D NMR spectroscopy. Similarly, in the 2D NOESY spectra of compound **11**, the cross peaks of the protons of the methyl group are observed both with the proton in position 7 and with the protons of the amino fragment in position 2. This indicates the presence of structure **11**.

Furthermore, 1-Methyl-4-nitrobenzotriazole (**12**) undergoes amination by TMHI in *t*-BuOK/DMSO to positions 7 and 5, to form 1-methyl-7-amino-4-nitrobenzotriazole (**13**) and 1-methyl-5-amino-4-nitrobenzotriazole (**14**) in a ratio of ~2:1, respectively [44,90,91] (Scheme 6, Table 1). The 2D NOESY spectra of compound **13** show the cross peak of the methyl protons at position 1 with NH$_2$ protons (at position 7). As for benzotriazole **14**, the

NOESY cross peaks of the methyl protons with H-7 and $NH_2$ protons with the H-6 proton are observed.

**Scheme 6.** VNS amination of 1-methyl-4-nitrobenzotriazole in a superbasic medium using 1,1,1-trimethylhydrazinium iodide.

**Table 1.** $^1H$, $^{13}C$ and $^{15}N$ NMR chemical shifts of nitroazoles and their aminated products (ppm) (DMSO-$d_6$).

| Compound | $\delta ^1H$ | $\delta ^{13}C$ | $\delta ^{15}N$ |
|---|---|---|---|
| **1** | 3.91 $CH_3$<br>8.22 s H-3<br>8.83 s H-5 | 39.71 $CH_3$<br>130.94 C-5<br>135.43 C-3<br>- C-4 | −18.3 $NO_2$<br>−69.9 N-2<br>−172.0 N-1 |
| **2** | 3.76 $CH_3$<br>7.80 s H-2<br>8.35 s H-5 | 34.18 $CH_3$<br>122.49 C-5<br>138.00 C-2<br>- C-4 | −18.1 $NO_2$<br>−127.7 N-3<br>−208.5 N-1 |
| **3** | 7.58 *p*-Ph<br>7.66 *m*-Ph<br>8.08 *o*-Ph<br>9.01 s H-5 | 119.37 *o*-Ph<br>129.81 *p*-Ph<br>130.05 *m*-Ph<br>132.99 C-5<br>138.22 *ipso*-Ph<br>154.22 C-4 | −27.0 $NO_2$<br>−50.8 N-1<br>−65.5 N-3<br>−122.8 N-2 |
| **4** | 3.92 s $CH_3$<br>7.80 d H-7<br>$^3J$ 9.0 Hz<br>8.18 d H-6<br>$^3J$ 9.0 Hz<br>8.49 s H-2<br>8.53 s H-4 | 31.20 $CH_3$<br>110.94 C-7<br>115.51 C-4<br>117.84 C-6<br>138.94 C-8<br>142.39 C-9<br>142.73 C-5<br>148.75 C-2 | −11.4 $NO_2$<br>−125.0 N-3<br>−220.9 N-1 |

**Table 1.** *Cont.*

| Compound | $\delta^1H$ | $\delta^{13}C$ | $\delta^{15}N$ |
|---|---|---|---|
| **5** | 3.96 s CH$_3$<br>7.53 d H-4<br>$^3J$ 8.9 Hz<br>7.80 d H-6<br>$^3J$ 8.9 Hz<br>8.34 s H-7<br>8.49 s H-2 | 33.29 CH$_3$<br>107.64 C-7<br>117.00 C-5<br>119.67 C-4<br>134.44 C-8<br>142.89 C-6<br>147.83 C-9<br>149.85 C-2 | −9.8 NO$_2$<br>−123.4 N-3<br>−217.1 N-1 |
| **6** | 3.56 s CH$_3$<br>7.38 br NH$_2$<br>7.84 s H-3 | 35.04 CH$_3$<br>117.84 C-4<br>134.37 C-5<br>146.01 C-3 | −18.5 NO$_2$<br>−92.1 N-2<br>−207.2 N-1<br>−316.9 NH$_2$ |
| **7** | 3.43 CH$_3$<br>7.23 s H-2<br>7.51 br NH$_2$ | 30.71 CH$_3$<br>124.0 br C-4<br>132.46 C-2<br>143.97 C-5 | −18.8 NO$_2$<br>−124.7 N-3<br>−228.5 N-1<br>−306.7 NH$_2$ |
| **8** | 4.50 br NH$_2$<br>7.25 *p*-Ph<br>7.46 *m*-Ph<br>7.54 *o*-Ph | 108.67 C-5<br>- C-4<br>117.58 *o*-Ph<br>126.48 *p*-Ph<br>129.70 *m*-Ph<br>142.75 *ipso*-Ph | −27.0 NO$_2$<br>−50.8 N-1<br>−65.5 N-3<br>−122.8 N-2<br>−309.5 NH$_2$ |
| **9** | 3.82 CH$_3$<br>6.88 d H-7<br>$^3J$ 9.3 Hz<br>7.65 br NH$_2$<br>7.90 d H-6<br>$^3J$ 9.3 Hz<br>8.18 s H-2 | 31.05 CH$_3$<br>99.92 C-7<br>120.38 C-6<br>124.90 C-5<br>131.67 C-9<br>137.46 C-8<br>140.50 C-4<br>143.80 C-2 | −3.3 NO$_2$<br>−131.8 N-3<br>−222.9 N-1<br>−307.6 NH$_2$ |
| **10** | 4.17 CH$_3$<br>6.92 d H-4<br>$^3J$ 9.2 Hz<br>7.46 br NH$_2$<br>7.84 d H-5<br>$^3J$ 9.2 Hz<br>8.21 s H-2 | 34.24 CH$_3$<br>109.37 C-4<br>119.98 C-5<br>122.56 C-8<br>126.57 C-6<br>137.10 C-7<br>148.44 C-2<br>148.76 C-9 | −3.2 NO$_2$<br>−129.0 N-3<br>−222.5 N-1<br>−305.6 NH$_2$ |

**Table 1.** *Cont.*

| Compound | $\delta\ ^1H$ | $\delta\ ^{13}C$ | $\delta\ ^{15}N$ |
|---|---|---|---|
| **11** | 3.58 CH$_3$<br>7.18 d H-4<br>$^3J$ 9.1 Hz<br>7.21 br NH$_2$<br>7.93 d H-5<br>$^3J$ 9.1 Hz<br>8.06 s H-7 | 28.84 CH$_3$<br>103.64 C-9<br>113.48 C-4<br>117.91 C-5<br>134.67 C-8<br>138.89 C-6<br>149.74 C-7<br>159.71 C-2 | −4.8 NO$_2$<br>−132.4 N-3<br>−224.5 N-1<br>−311.4 NH$_2$ |
| **12**<br>This work | 4.42 s CH$_3$<br>7.79 dd H-6<br>$^3J$ 8.3 Hz<br>$^3J$ 7.7 Hz<br>8.32 d H-5<br>$^3J$ 7.7 Hz<br>8.39 d H-7<br>$^3J$ 8.3 Hz | 34.98 CH$_3$<br>118.83 C-5<br>121.52 C-7<br>126.78 C-6<br>135.86 C-8<br>137.41 C-9<br>137.86 br C-4 | 8.3 N-2<br>−10.0 NO$_2$<br>−42.3 N-3<br>−153.7 N-1 |
| **13**<br>This work | 4.51 s CH$_3$<br>6.59 d H-6<br>$^3J$ 8.8 Hz<br>7.21 NH$_2$<br>8.12 d H-5<br>$^3J$ 8.8 Hz | 38.12 CH$_3$<br>109.80 C-6<br>123.94 C-8<br>126.82 C-5<br>135.86 C-4<br>141.21 C-7<br>142.60 C-9 | 8.9 N-2<br>−9.1 NO$_2$<br>−44.3 N-3<br>−159.5 N-1<br>−312.40 NH$_2$ |
| **14**<br>This work | 4.22 s CH$_3$<br>6.21 d H-6<br>$^3J$ 9.0 Hz<br>7.92 d H-7<br>$^3J$ 9.0 Hz<br>8.25 br NH$_2$ | 35.30 CH$_3$<br>99.08 C-7<br>125.43 C-5<br>126.09 C-6<br>137.39 br C-4<br>137.70 C-8<br>141.32 C-9 | 9.3 N-2<br>−11.5 NO$_2$<br>−49.0 N-3<br>−157.9 N-1<br>−309.1 NH$_2$ |
| **15**<br>[95] | 4.03 s CH$_3$<br>7.13 s H-3<br>7.58 *p* Ph<br>7.66 *m* Ph<br>8.08 *o* Ph<br>8.21 br NH$_2$<br>8.79 s H-5 | | −8.1 4-NO$_2$<br>−9.8 6-NO$_2$<br>−215.5 N-3<br>−318.8 N-2<br>This work |
| **18a**<br>This work | 7.46 br 7NH$_2$<br>7.65 br 4NH$_2$<br>8.06 s H-2 | 102.24 C-7<br>125.15 C-4<br>125.64 C-8<br>130.18 C-9<br>138.34 C-6<br>142.05 C-2<br>149.23 C-5 | −7.3 NO$_2$<br>−192.4 N-1.3<br>−301.4 NH$_2$ |

**Table 1.** *Cont.*

| Compound | $\delta^1H$ | $\delta^{13}C$ | $\delta^{15}N$ |
|---|---|---|---|
| **19** | 6.36 br $NH_2$<br>8.47 s H-3,5 | 144.51 C-3,5 | −66.1 N-1,2<br>−197.2 N-4<br>−314.3 $NH_2$ |
| **20** | 3.46 s $CH_3$<br>7.12 s H-3<br>8.51 s H-3′,5′<br>14.1 NH | 34.02 $CH_3$<br>128.39 C-3<br>146.20 C-4<br>155.04 C-5 | −41.0 $NO_2$<br>−64.3 N-1′,2′<br>−117.0 N-2<br>−180.8 N-4′<br>−220.5 N-1<br>−305.5 NH |
| **21** | 3.98 s $CH_3$<br>7.81 d H-6<br>$^3J$ 8.6 Hz<br>8.00 d H-7<br>$^3J$ 8.6 Hz<br>8.35 s H-2 | 31.78 $CH_3$<br>110.18 C-6<br>121.26 C-7<br>127.10 C-9<br>133.92 C-8<br>143.92 C-4<br>144.02 C-2<br>149.11 C-5 | +29.9 NO<br>+30.6 NO<br>−130.6 N-3<br>−220.9 N-1 |
| **22** | 3.58 s $CH_3$<br>7.18 d H-4<br>$^3J$ 8.7 Hz<br>7.93 d H-5<br>$^3J$ 8.7 Hz<br>8.30 s H-2 | 28.84 $CH_3$<br>103.64 C-9<br>113.48 C-4<br>117.91 C-5<br>134.67 C-8<br>138.89 C-6<br>149.74 C-7<br>159.71 C-2 | +30.7 NO<br>+31.9 NO<br>−135.6 N-3<br>−224.7 N-1 |

Likewise, 2-Aryl-1-methyl-4,6-dinitroindoles are aminated by TMHI in the presence of *t*-BuOK/*DMSO* exclusively at position 7 to afford 7-amino-substituted, *ortho*-nitro amines (Scheme 7, Table 1) [95]. The transformation of 7-$NH_2$ allows the employment of 2-aryl-4,6-dinitroindoles as a basis to obtain new indole derivatives with potentially useful biological properties.

**Scheme 7.** VNS amination of nitroindoles in a superbasic medium using 1,1,1-trimethylhydrazinium iodide.

As is known, in 4-nitro-and 6-nitroindazoles, VNS takes place at the C-7 atom. [35,44,96]. The structure of the reaction product was refined by X-ray crystallographic analysis (Scheme 8) [97].

**Scheme 8.** Vicarious nucleophilic substitution of hydrogen in nitroindazoles in a superbasic medium.

The *C*-amination of 1-methyl-3-nitropyrazole (**16**), 2-methyl-4-nitroimidazole and 1,2-dimethyl-4-nitroimidazole does not occur under the same conditions: the corresponding starting compounds are recovered from the reaction in yields of up to 75%. Thus, 1-methyl-3-nitropyrazole (**16**), unlike its 1-methyl-4-nitropyrazole (**1**) isomer, does not interact with 1,1,1-trimethylhydrazinium halides under the above reaction conditions [82].

In order to explain such a significant difference in the behavior of these isomers under the conditions of VNS amination, we performed ab initio calculations for the B3LYP/6-31G of 1-methyl-3-nitro- (**16**) and 1-methyl-4-nitropyrazole (**1**) [82]. The calculations show significant differences in the charge density on the carbon atoms of these compounds, which is in good agreement with the observed differences in their reactivity (Figure 1).

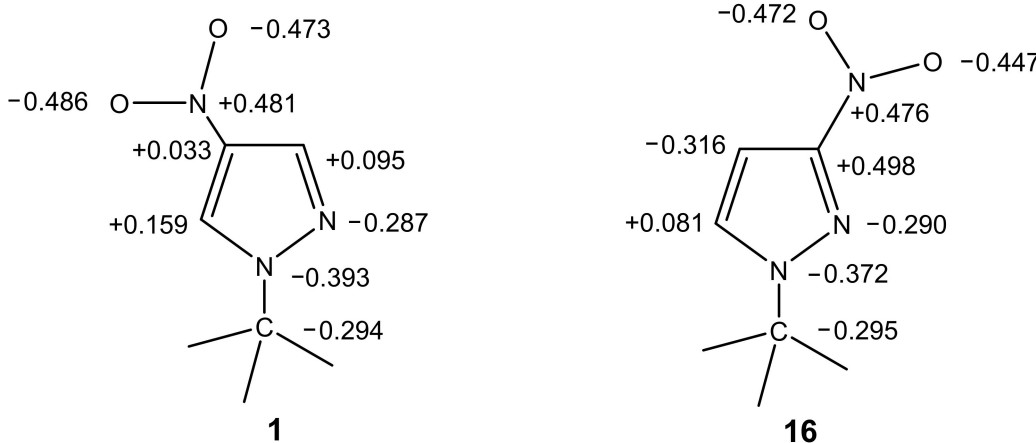

**Figure 1.** The charges at the atoms in the molecules of 1-methyl-4-nitropyrazole and 1-methyl-3-nitropyrazole according to ab initio B3LYP/6-31G calculations.

The largest positive charge in 1-methyl-4-nitropyrazole (**1**) is concentrated on the C-5 atom, and is significantly higher than the charge on the C-3 atom. This is in good agreement with the observed *C*-amination of 1-methyl-4-nitropyrazole at position 5. At the same time, according to the calculated data, nucleophilic attack in 1-methyl-3-nitropyrazole (**16**) can occur only at the C-5 atom, although the nucleophilicity of this substrate is substantially lower than that of the 1-methyl-4-nitro isomer. This probably also explains our unsuccessful attempts to aminate 1-methyl-3-nitropyrazole [82]. However, despite the simplicity of such an explanation, the analysis of the charge distribution does not completely clarify the mechanism of the VNS, because in some cases the reactions proceed despite this factor.

The authors [84] succeeded in the amination of 3,5-dinitropyrazole (**17**) with 1,1,1-trimethylhydrazinium iodide in a superbasic medium with the formation of 4-amino-3,5-dinitropyrazole (**18**) (Scheme 9). Apparently, two electron-withdrawing nitro groups redistribute the negative charge in the molecule, which ensures the entry of the amino group into position 4.

**Scheme 9.** VNS amination of 3,5-dinitropyrazole in a superbasic medium using 1,1,1-trimethylhydrazinium iodide.

The reaction proceeds very quickly at room temperature with a good yield (70%), but—as shown by X-ray structural analysis—in fact, a crystalline solvate of product **18** with DMSO of the composition 1:1 is formed. Its recrystallization from acetonitrile, surprisingly, gives the same solvate with DMSO, while recrystallization from water leads to the monohydrate. After some experimentation, the authors [84] found that the recrystallization of the monohydrate from butyl acetate with heptane leads to a pure crystalline product, 4-amino-3,5-dinitropyrazole (**18**).

Similarly, in the framework of this work, we carried out the amination of 5,6-dinitrobenzimidazole under the conditions of the VNS technique, and found that the reaction leads to the formation of 4,7-diamino-5,6-dinitrobenzimidazole (**18a**) (Table 1).

### 2.2. The Products of the Interaction of N-Substituted Nitroazoles with 4-amino-1,2,4-triazole

As a result of the reaction of 1-methyl-4-nitropyrazole (**1**) with 4-amino-1,2,4-triazole (**19**), along with amination product **6**, (1-methyl-4-nitropyrazol-5-yl) (1,2,4-triazol-4-yl) amine (**20**) is formed (Scheme 10, Table 1) [44,81,83]. It should be noted that isomeric 1-methyl-3-nitropyrazole does not interact with 4-amino-1,2,4-triazole.

**Scheme 10.** VNS amination of 1-метил-4-nitropyrazole with 4-amino-1,2,4-triazole.

In the $^1$H NMR spectrum of **20**, in addition to the signals of the protons of the methyl group and H-3 in a low field, a singlet (8.51 ppm) appears, which relates to the equivalent protons of the triazole ring, and a broad signal of the NH-proton. The introduction of an amino group into the pyrazole ring (**1**→**6**) leads to a significant shift of the resonance signals of nitrogen atoms N-1 and N-2 in a high field (on average by 30 ppm), which may indicate an increase in the π-electron density in an aromatic system [44] and, naturally, the π-donor character of the NH$_2$-group.

A high field shift of the $^{15}$N NMR signals of the N-1 and N-2 atoms (by 13.3 and 25 ppm, respectively) is also observed for (1-methyl-4-nitropyrazol-5-yl)(1,2,4-triazole- 4-yl) amine (**20**) in comparison with the characteristic for **6**. As can be seen from Table 1, the value of the chemical shift of the nitrogen-15 atom of the nitro group is insensitive to the introduction of the amino group into the azole ring (~0.2 ÷ 2 ppm) (Table 1). Nevertheless, the position of the resonance signal of the nitrogen-15 atom of the nitro group of compound **20**, in comparison with **6**, is shifted in a high field by more than 20 ppm. Apparently, the screening of the nitrogen nucleus of the nitro group in **20** increases due to the formation of a hydrogen bond of the N-H ... O-N type (Figure 2):

**Figure 2.** Possible hydrogen bonding in (1-methyl-4-nitropyrazol-5-yl)(1,2,4-triazole-4-yl)amine (**20**).

In addition, the formation of a hydrogen bond in (1-methyl-4-nitropyrazol-5-yl) (1,2,4-triazol-4-yl) amine (**20**) is also evidenced from the rather large chemical shift of the NH proton (14.1 ppm).

A similar shift (10–20 ppm) of the nitrogen atom-14 signal of the oxide group in 4-substituted 2-methyl-5-nitro-1,2,3-triazole-1-oxides, as compared to unsubstituted ones, was attributed [98] to the formation of an intramolecular hydrogen bond.

The reaction of 1-methyl-4-nitroimidazole (**2**) with 4-amino-1,2,4-triazole (**19**) under similar conditions gives exclusively the amination product, 5-amino-1-methyl-4-nitroimidazole (**7**), and 1,2,4-triazole as a by-product.

The reaction of the *C*-amination of 1-methyl-5-nitrobenzimidazole (**4**) and 1-methyl-6-nitrobenzimidazole (**5**) with 4-amino-1,2,4-triazole (**19**) in the system DMSO/*t*-BuOK at room temperature leads to 1-methyl-[4,5]-furazanobenzimidazole (**21**) and 1-methyl-[6,7]-furazanobenzimidazole (**22**), together with the expected amination products, 1-methyl-4-amino-5-nitro- (**9**) and 1-methyl-4-amino-6-nitrobenzimidazole (**10**) [44,90,91] (Schemes 11 and 12, Table 1). In addition, the presence of $Cu_2Cl_2$ or $AgNO_3$ catalysts increases essentially the yield of products. The $^1$H NMR technique was applied to monitor the reaction for some days. It has been shown that the increase of the reaction time (up to 14 days) leads to the predominant formation of furazanobenzimidazoles **21** and **22** (ratio of **1:2**). The structure of the reaction products was proved by multinuclear NMR ($^1$H, $^{13}$C, $^{15}$N), 2D NOESY NMR techniques and elemental analysis.

**Scheme 11.** VNS amination of 1-methyl-5-nitrobenzimidazole (**4**) with 4-amino-1,2,4-triazole.

**Scheme 12.** VNS amination of 1-methyl-6-nitrobenzimidazole (**5**) with 4-amino-1,2,4-triazole.

The $^{15}$N NMR spectra of compounds **21** and **22** contain 4 signals, two of which at $-(220–224)$ and $-(130–135)$ ppm belong, respectively, to the pyrrole (N-1) and pyridine (N-3) nitrogen atoms of the imidazole ring, and the other two, located in a significantly higher

field +(29–32) ppm, refer to the nitrogen atoms of the NO fragments, and are characteristic of the furazan cycles [41,99]. The $^1$H NMR spectra show the high-field signals of the methyl group protons, a low-field singlet signal (8.35 ppm) related to the proton in position 2, and two doublets belonging to H-6 and H-7 (H-4 and H-5). Based on the 2D NOESY spectrum, it can be shown that the signal at 8.00 ppm in furazanobenzimidazole **21** refers to the proton in position 7. The assignment of $^{13}$C NMR signals was made on the basis of two-dimensional spectra HSQC-GP $^1$H-$^{13}$C (Heteronuclear Single Quantum Correlation Gradient Pulse) and HMBC-GP $^1$H-$^{13}$C (Heteronuclear Multiple Bond Correlation Gradient Pulse).

The vicarious nucleophilic substitution of hydrogen in heterocyclic compounds is a convenient—and in some cases the only—method for introducing various functional groups and substituents—in particular, an amino group—into activated heterocyclic systems. Thus, VNS is a key step in the synthesis of, for example, purine bases from commercially available nitroimidazoles. It can be said with certainty that, in the near future, the VNS of hydrogen will occupy a worthy place in the synthetic arsenal of researchers, and will expand our understanding of the subtle processes that occur during the synthesis of organic compounds.

### 2.3. The Structure of the Reaction Products of Nitrobenzene with 1,1,1 trimethylhydrazinium Halides, and 4-amino-1,2,4-triazole

Instead of the expected mixture of *ortho-* and *para*-nitroanilines, as reported in [75], we found that the reaction of nitrobenzene (**23**) with 1,1,1-trimethylhydrazinium halides in an absolute DMSO medium in the presence of *t*-BuOK at 20 °C leads to the formation of not only *para*-nitroaniline (**24**) but also bis(*para*-nitrophenyl)amine (**25**) (Scheme 13) [81,85]:

**Scheme 13.** VNS amination of nitrobenzene (**23**) with 4-amino-1,2,4-triazole (**19**).

In this case, the corresponding *ortho*-isomer is not formed, and the nature of the used halide anion (Cl, Br, I) does not significantly affect the yield of the final products. The $^1$H NMR spectrum of compound **25** contains a singlet of the NH-proton (9.92 ppm) and two doublets of two pairs of equivalent protons of the benzene rings at 7.35 and 8.20 ppm with a constant $^3J$ = 9.5 Hz. Its $^{15}$N NMR spectrum exhibits signals that are characteristic [99] of this type of compounds, and which relate to the nitrogen atom of two equivalent nitro groups (−10.7 ppm) and the nitrogen atom of the amino group (−271.8 ppm) [85].

At the same time, the interaction of 4-amino-1,2,4-triazole (**19**) with nitrobenzene (**23**) under the same conditions leads to the formation of several products, the amination product **24** and the condensation products bis(*para*-nitrophenyl) amine (**25**), (*para*-nitrophenyl) (1,2,4-triazol-4-yl) amine (**26**) and bis(*para*-nitrophenyl) (1,2,4-triazol-4-yl) amine (**27**) (Scheme 14).

The ratio of products of **24**, **25**, **26**, and **27** depends on the conditions of the nucleophilic substitution reaction (temperature, reaction time, and the ratio of reagents). For example, when the reaction mixture was heated for four hours, *para*-nitroaniline (**24**) and (*para*-nitrophenyl)(1,2,4-triazol-4-yl) amine (**26**) were isolated (Table 2).

**Scheme 14.** Products of the interaction of nitrobenzene (**23**) with 4-amino-1,2,4-triazole (**19**) in a superbasic medium.

**Table 2.** $^1$H and $^{13}$C NMR chemical shifts of nitrobenzene amination products (ppm) (DMSO-$d_6$).

| | Compound | δ $^1$H | δ $^{13}$C |
|---|---|---|---|
| **24** |  | 6.61 d H-3,5 <br> $^3J$ = 8.3 Hz <br> 8.05 d H-2,6 <br> $^3J$ = 8.3 Hz <br> 6.70 br NH$_2$ | 111.45 C-3,5 126.44 C-2,6 <br> 136.00 C-1 156.00 C-4 |
| **25** |  | 7.35 d H-3,5 <br> $^3J$ = 9.5 Hz <br> 8.20 d H-2,6 <br> $^3J$ = 9.5 Hz | 117.04 C-3,5 125.77 C-2,6 <br> 140.56 C-1 147.60 C-4 |
| **26** |  | 6.59 d H-3,5 <br> $^3J$ = 9.2 Hz <br> 8.16 d H-2,6 <br> $^3J$ = 9.2 Hz <br> 8.87 s H-3′,5′ <br> 10.52 br NH | 111.70 C-3,5 126.13 C-2,6 <br> 140.05 C-1 144.08 C-3′,5′ <br> 156.00 C-4 |
| **27** |  | 6.79 d H-3,5 <br> $^3J$ = 9.0 Hz <br> 7.44 d H-2,6 <br> $^3J$ = 9.0 Hz <br> 8.61 s H-3′,5′ | 113.22 C-3,5 129.42 C-2,6 <br> 140.95 C-1 142.67 C-3′,5′ <br> 158.08 C-4 |

For bis(*para*-nitrophenyl) amine (**25**), which was also obtained by counter-synthesis, it was possible to measure the NMR spectra of the nitrogen-15 nucleus (Figure 3).

$$\delta^{15}\text{N, ppm}$$

**Figure 3.** $^{15}$N NMR chemical shifts of bis(*para*-nitrophenyl) amine (**25**) (ppm) (DMSO-$d_6$).

In order to prove these structures, multinuclear and 2D NMR spectroscopy techniques were used.

Thus, the results of the study of the *C*-amination of nitroazoles via the vicarious nucleophilic substitution of hydrogen using multinuclear NMR spectroscopy showed that this reaction proceeds mainly with *N*-substituted nitroazoles, with the exception of some examples indicated above. All of the attempts to carry out the amination of azoles with an acid NH-proton (nitropyrazole, nitroimidazole, etc.) lead to the isolation of the starting compound. This can be explained by the deprotonation of the substrate in a superbasic medium. In this case, the interaction of the formed negatively charged anion with the nucleophile is hindered.

## 3. Organylpyrazole Derivatives

The development of preparative methods for the synthesis of pyrazoles, especially halopyrazoles, from available products and the establishment of their structure is an urgent task. The known methods for obtaining these practically important substances are complex and multi-stage, while the yields of the final products are low [100,101]. As a result of the search for new ways of recycling rocket fuel, as mentioned above, we have proposed an original method for the formation of functionally substituted pyrazoles or isoxazoles using heptyl (or hydroxylamine) or other reagents [102–112].

The reaction mechanism apparently involves the initial formation of the 2-chlorovinyl ketone dimethylhydrazone, followed by an intramolecular attack of the nucleophilic dimethylamine fragment of the β-carbon atom of the vinyl group (Scheme 15). We have established the structure of 1-methyl-3-alkyl-, -aryl-, -chloroalkyl- and -perfluoroalkyl-5-chloropyrazoles (**28**) formed by the reaction of the corresponding 2,2-dichlorovinyl ketones with 1,1-dimethylhydrazine [102,103].

**Scheme 15.** Formation of 3-substituted 1-methyl-5-chloropyrazole in the reaction of dichlorovinyl ketones with 1,1-dimethylhydrazine.

The resulting *N,N*-dimethylpyrazolinium chloride is dequaternized with the release of the target aromatic pyrazole and MeCl, which reacts with 1,1-dimethylhydrazine to form 1,1,1-trimethylhydrazinium chloride [102,103]. The structure of new 1-methyl-3-organyl-5-halopyrazoles (**28**) was established by multinuclear NMR ($^1$H, $^{13}$C, $^{15}$N, and $^{19}$F), and the effect of the nature of the substituent at position 3 on the parameters of the NMR spectra was studied (Table 3) [103]. In the $^1$H NMR spectra of pyrazole **28**, the resonance signals of the H-4 protons appear in the region of 5–7 ppm. This signal shifts downfield when passing from alkyl substituents at position 3 (**28a**–**28d**) (5.6–6.0) to electron-withdrawing ($CF_3$, $CH_2Cl$) (6.2–6.4) or aromatic (Ar) (6.4–6.9).

**Table 3.** $^1$H, $^{13}$C and $^{15}$N NMR chemical shifts of 3-substituted 1-methyl-5-chloropyrazole **28** (ppm, $CDCl_3$).

| Compd | R | δ $^1$H | | δ $^{13}$C | | | | δ $^{15}$N | |
|---|---|---|---|---|---|---|---|---|---|
| | | H-4 | $CH_3$ | C-3 | C-4 | C-5 | $CH_3$ | N-1 | N-2 |
| **28a** | $CH_3$ | 5.94 | 3.72 | 148.00 | 103.77 | 127.03 | 35.53 | | |
| **28b** | $C_2H_5$ | 5.62 | 3.54 | | | | | | |
| **28c** | $C_3H_7$ | 5.97 | 3.75 | 152.95 | 103.10 | 127.10 | 35.84 | −188.2 | −78.4 |
| **28d** | $i$-$C_3H_7$ | 5.99 | 3.73 | | | | | | |
| **28e** | $CH_2Cl$ | 6.24 | 3.78 | 118.40 | 103.97 | 127.86 | 38.61 | −183.8 | −74.4 |
| **28f** [a] | $CF_3$ | 6.43 | 3.84 | 141.80 | 103.40 | 128.89 | 36.78 | −178.5 | −74.3 |
| **28g** | $C_6H_5$ | 6.44 | 3.81 | | 101.84 | 128.00 | | | |
| **28h** | $4$-$CH_3C_6H_4$ | 6.44 | 3.85 | 150.96 | 101.63 | 128.09 | 36.30 | −184.8 | −81.6 |
| **28i** | $4$-$CH_3OC_6H_4$ | 6.39 | 3.79 | 150.75 | 101.30 | 128.05 | 36.19 | −185.3 | −82.8 |
| **28j** | $4$-$BrC_6H_4$ | 6.44 | 3.85 | 149.54 | 101.65 | 128.25 | 36.24 | −183.2 | −80.3 |
| **28k** | $4$-$ClC_6H_4$ | 6.39 | 3.81 | 149.81 | 101.86 | 128.00 | 36.42 | −184.8 | −81.6 |
| **28l** [b] | $4$-$NO_2C_6H_4$ | 6.58 | 3.90 | 148.45 | 102.83 | 129.11 | 36.66 | −180.0 | −76.8 |
| **28m** [c] | $3$-$NO_2C_6H_4$ | 6.58 | 3.90 | 148.21 | 102.01 | 129.43 | 36.33 | −181.3 | −78.0 |

[a] $\delta^{19}$F $= -63.4$, $\delta^{13}$C($CF_3$) $= 121.69$, $^1J_{C-F} = 269.1$ Hz, $^2J_{C-F} = 38.7$ Hz; [b] $\delta^{15}$N($NO_2$) $= -11.2$ ppm; [c] $\delta^{15}$N($NO_2$) $= -11.1$ ppm.

The protons of the methyl group are less sensitive to the influence of the substituent. The position of the C-4 and C-5 resonance signals in the $^{13}$C NMR spectra of these compounds is also not very sensitive to substituent change ($\Delta\delta^{13}$C ~ 3 ppm), and is close to that observed for 1-methylpyrazole (105.7 and 128.7 ppm, respectively) [44].

As already discussed, in N-substituted azoles (which are incapable of tautomeric rearrangements) the screening of the "pyrrole" nitrogen atom is much higher than that of the "pyridine" one, which makes it possible to unambiguously assign signals in the $^{15}$N NMR spectra of this series of compounds. The resonance signals of nitrogen-15 atoms of nitro groups **28l** and **28m** (~−11 ppm) lie in the region characteristic of the nitrogen resonance of $^{15}$N aromatic and heterocyclic nitro compounds (Table 3) [35,45].

The $^{15}$N chemical shifts of the N-1 atom are in the range of −178 ÷ −188 ppm, and those of the N-2 atom are by about 100 ppm more (−76 ÷ −81 ppm). With an increase in the electron-withdrawing properties of the substituent (R), the resonance signal of the $^{15}$N "pyrrole" nitrogen atom shifts downfield (~10 ppm), while the dependence of the δ $^{15}$N values of the "pyridine" nitrogen atom on the nature of the substituent is more complex (Table 3) [103].

In conclusion, the discovered reaction makes it possible to efficiently access 3-substituted 1-methyl-5-chloropyrazole, a precursor of promising multifunctional pyrazole derivatives which is of interest in terms of biological activity and drug and pesticide design. The structure of the obtained compounds was proven by the methods of two-dimensional and multinuclear NMR spectroscopy.

## 4. Functional 1,2,3-Triazole Derivatives

Furthermore, 1,2,3-Triazoles play a significant role in the chemistry of heterocycles, and are widely used in biological and medicinal chemistry. Functionalized triazoles are

used as high-energy materials, ionic liquids, dyes, universal bases in peptide nucleic acids, synthons for fine organic synthesis, and precursors for nanomaterials [9,11,13]. Such a various use of 1,2,3-triazoles requires an understanding of the features of their electronic and stereochemical structure, spectral properties, and tautomeric transitions [113–116].

The stereochemical behavior and annular ptototropic tautomerism of 4-substituted 1,2,3-triazoles was studied by multinuclear [1]H, [13]C, [15]N and [29]Si NMR spectroscopy and quantum chemistry [113,114,117,118].

Likewise, 4-Substituted oximes of 1,2,3-triazole-5-carbaldehyde were synthesized by the interaction of corresponding propinals, azide, and hydroxylamine in an aqueous methanol solution with microwave irradiation (Scheme 16) [114,115]:

R = Me₃Si, Et₃Si, Me₃C

**Scheme 16.** Formation of 4-substituted 1,2,3-triazole-5-carbaldehyde oximes.

It is known that 1,2,3-triazoles in solution exist as an equilibrium mixture of three tautomers: **A**, **B**, and **C**. This produces some experimental difficulties in assigning NMR chemical shifts and determining their structure (Scheme 17) [35,45,47].

**A**                                             **B**                                             **C**

**Scheme 17.** Tautomeric equilibrium of 1,2,3-triazole-5-carbaldehyde oximes.

The annular prototropy of azoles, and in particular 1,2,3-triazoles, is a relatively rapid process in the timescale of NMR. Consequently, the [15]N NMR method fails to detect the signals of all of the triazolyl nitrogen atoms in the molecule of the equilibrium tautomeric mixture of 1,2,3-triazole. The two-dimensional [15]N NMR spectra HMBC {1H-15N} of 4-(trimethylsilyl)-1,2,3-triazole-5-carbaldehyde oxime show two cross peaks from two nitrogen atoms with the proton of the CH = N group at −5.6 and −27.9 ppm (Table 4) [113].

The [1]H NMR spectra of compounds **29–31**, show broad OH and NH signals in the regions of 11 and 15 ppm, respectively. The carbon signals in the [13]C NMR spectra are also broadened, which indicates the existence of a prototropic exchange process. The assignment of carbon signals was made using two-dimensional {1H-13C} HMBC NMR spectra. Cross-peaks of methyl protons with the carbon atom in 4-(trimethylsilyl)-1,2,3-triazole-5-carbaldehyde oxime (**29**) are observed at 132.1 ppm (C-4), whereas the cross-peaks of the CH = N proton with carbon atoms are at 132.1 ppm (C-4) and 146.6 ppm (C-5) (Table 4). The orientation of the oxime fragment is defined from the carbon spectrum [13]C(H) without proton decoupling. The constant values $^1J_{C6-H}$ in the [13]C NMR spectra of **29–31** are 164–166 Hz, which indicates the existence of the molecules in the form of the *E*-isomer (Scheme 18), as shown in [119,120].

**Table 4.** $^1$H, $^{13}$C and $^{15}$N NMR chemical shifts of 4-substituted 1,2,3-triazole-5-carbaldehyde oxime (**29**, **30**, **31**) (ppm, DMSO-$d_6$).

| | Structure | $^1$H | $^{13}$C | $^{15}$N * |
|---|---|---|---|---|
| **29** | | 0.39 SiMe$_3$<br>8.21 CH=N<br>11.30 br OH<br>15.1 br NH | -0.50 SiMe$_3$<br>132.7 C-4<br>142.5 CH=N<br>$^1J_{CH}$ = 164.0 Hz<br>147.3 C-5 | −5.6 N-7<br>−27.9 N-1 |
| **30** | | 0.87 Me$_3$<br>0.97 SiCH$_2$<br>8.20 CH=N<br>11.24 br OH<br>15.08 br NH | 3.84 SiCH$_2$<br>7.06 Me$_3$<br>129.03 C-4<br>141.33 CH=N<br>$^1J_{CH}$ = 164.7 Hz<br>146.68 C-5 | −5.8 N-7<br>−7.7 N-2<br>−120.0 N-3<br>$^1J_{NH}$ = 101.6 Hz |
| **31** | | 1.35 Me$_3$<br>8.25 CH=N<br>11.40 br OH<br>14.75 br NH | 28.98 CH$_3$<br>59.74 C- CH$_3$<br>136.93 C-5<br>141.68 CH=N<br>$^1J_{C-H}$ = 165.8 Hz<br>153.09 C-4 | −3.1 N-7<br>−48.5 N-1<br>−132.0 N-3<br>$^1J_{NH}$ = 114.2 Hz |

* Assignment made based on the calculated $^{15}$N NMR chemical shift values (see below).

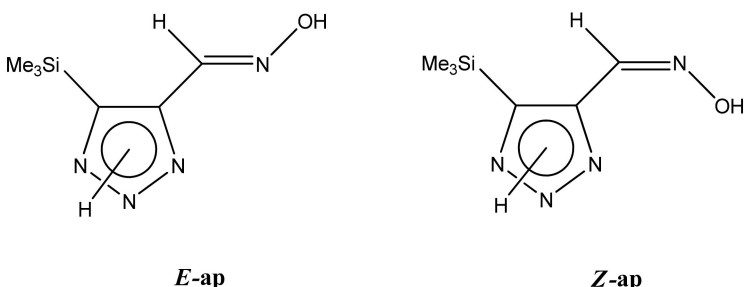

**Scheme 18.** *E*-, *Z*-anti-periplanar conformations of 4-(trimethylsilyl)-1,2,3-triazole-5-carbaldehyde oxime (**29**).

The prevalence of the E-conformation was also shown by theoretical calculations of the energy barrier, which was 2.10, 1.40, and 3.40 kcal/mol for 1*H*-, 2*H*-, and 3*H*-triazoles, respectively.

The fraction of each possible tautomer **A**, **B** and **C** was determined using quantum-chemical calculations. It is not excluded that two *E*-conformers can exist in an exchange equilibrium (Scheme 19) [113]:

The geometric parameters of *E*-conformers were optimized at the MP2/aug-cc-pVTZ level. The conformers ***E*-sp** are more preferable than ***E*-ap**, with an energy difference of 3.91 and 5.20 kcal/mol for 2*H*- and 3*H*-triazoles, respectively. The distance between the N-7 atom and the nearest methyl protons of the trimethylsilyl group in ***E*-sp** is about 2.7 Å, which is higher than the sum of their Van der Waals radii (Figure 4). There are no obvious prerequisites for the formation of intramolecular hydrogen bonds. The energy of this tetrel bond was estimated to be ca. −1.07 kcal/mol. Thus, there is no denying the presence of an intramolecular interaction between nitrogen N-7 and silicon atoms, which leads to stabilization of the syn-periplanar form [113].

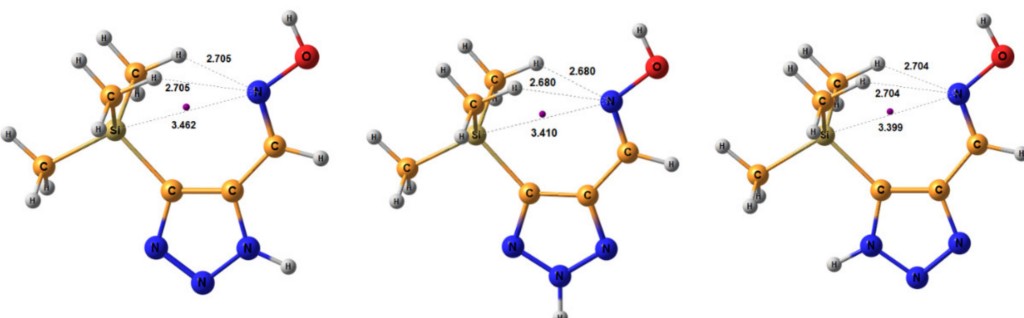

**Scheme 19.** *E*-anti- and syn-periplanar conformers of **29** oxime.

**Figure 4.** Equilibrium geometries of the *E*- syn-periplanar conformers of 4-(trimethylsilyl)-1,2,3-triazole-5-carbaldehyde oxime, optimized at the MP2/aug-cc-pVTZ level. The interatomic distances are given in Å. Violet corresponds to the bond's critical point (3, −1).

The conformational analysis of the studied 1,2,3-triazole oxime indicates that the *E*-syn-periplanar form is energetically dominant compared to the *Z*-syn-periplanar form (Table 5).

**Table 5.** Possible conformers of 4-trimethylsilyl-1,2,3-triazole-5-carbaldehyde oxime, optimized at the MP2/aug-cc-pVTZ level [113].

| Conformer | Structure | *E*, Hartree | Δ*E*, kcal/mol * |
|---|---|---|---|
| *E-ap* | | −818.1342766 | 4.71 |
| | | −818.1355085 | 3.91 |
| | | −818.1316053 | 6.45 |

**Table 5.** *Cont.*

| Conformer | Structure | *E*, Hartree | Δ*E*, kcal/mol * |
|---|---|---|---|
| *Z-ap* |  | −818.1375023 | 2.61 |
| |  | −818.1376553 | 2.51 |
| |  | −818.1368360 | 3.05 |
| *E-sp* |  | −818.1317761 | 6.34 |
| |  | −818.1415116 | 0 |
| |  | −818.1395942 | 1.25 |

**Table 5.** *Cont.*

| Conformer | Structure | *E*, Hartree | Δ*E*, kcal/mol * |
|---|---|---|---|
|  |  | −818.1278248 | 8.91 |
| *Z-sp* |  | −818.1362968 | 3.40 |
|  |  | −818.1363759 | 3.34 |

* relative to the syn-periplanar form of (*E*)-4-trimethylsilyl-2*H*-1,2,3-triazole-5-carbaldehyde oxime.

The ratio of tautomers (**A**, **B**, and **C**) was determined by comparing the experimental and calculated $^1$H, $^{13}$C, and $^{15}$N NMR chemical shifts of individual tautomers. The values of chemical shifts N-1, N-7, C-4, and C-5 can serve as reference values (Table 6). The comparison of their values indicates a substantial predominance (79.3%) of the tautomeric form **C** in the equilibrium mixture, while the proportions of tautomers **A** and **B** are 8.7% and 12.0%, respectively [113].

The optimization of the geometry of the studied compounds, their tautomers and conformers was performed at the MP2/aug-cc-pV5Z level, taking into account solvent effects within the Integral Equation Formalism Polarizable Continuum Model (IEF-PCM) [121]. All of the calculations of $^1$H, $^{13}$C and $^{15}$N NMR isotropic magnetic shielding constants were obtained with the CFOUR package [122] at the CCSD level, in combination with Jensen's triple-split basis set, pcS-2, specially optimized for the calculation of NMR chemical shifts [123].

Thus, the data of the quantum chemical calculations and NMR studies showed that 4-trimethylsilyl-1,2,3-triazole-5-carbaldehyde oxime exists in solution as the *E*-isomer. In addition, the calculation results indicate the complete prevalence of the conformer *E*-sp (synperiplanar form). The $^{15}$N NMR signals of the above compounds have been unambiguously assigned by the comparison of the experimental and calculated (at the CCSD/pcS-2 level) NMR chemical shifts (see Tables 4 and 5). As such, using the approach proposed, it can be found that the tautomer of 4-trimethylsilyl-**3*H***-1,2,3-triazole-5-carbaldehyde oxime is predominant in a solution in an equilibrium mixture.

**Table 6.** Calculated $^1$H, $^{13}$C and $^{15}$N NMR chemical shifts of tautomers **A**, **B**, and **C** (ppm) (MP2/aug-cc-pVTZ level).

| Structure | | $^1$H | | $^{13}$C | | $^{15}$N | |
|---|---|---|---|---|---|---|---|
| **A** |  | NH | 11.37 | CH$_3$ | 13.5 | N-1 | −149.2 |
| | | OH | 7.84 | C-4 | 147.3 | N-2 | 13.4 |
| | | N=CH | 7.12 | C-5 | 141.7 | N-3 | 18.9 |
| | | CH$_3$ | −0.70 | N=CH | 139.1 | N-7 | 38.3 |
| **B** |  | NH | 11.48 | CH$_3$ | 0.2 | N-1 | −48.8 |
| | | OH | 7.06 | C-4 | 150.9 | N-2 | −130.0 |
| | | N=CH | 8.23 | C-5 | 151.1 | N-3 | −31.9 |
| | | CH$_3$ | 0.30 | N=CH | 141.1 | N-7 | −3.4 |
| **C** |  | NH | 11.15 | CH$_3$ | −0.3 | N-1 | −8.6 |
| | | OH | 7.05 | C-4 | 135.3 | N-2 | 6.8 |
| | | N=CH | 8.17 | C-5 | 150.3 | N-3 | −143.2 |
| | | CH$_3$ | 0.30 | N=CH | 140.3 | N-7 | −5.4 |

## 5. Functional Thiazole Derivatives

### 5.1. The Structure of 2,4-disubstituted Thiazoles

The reactions of organylthiochloroacetylenes with *S,N*-containing bifunctional nucleophiles are ambiguous and, depending on the nature of the nucleophile and experimental conditions, lead to different products. As such, (alkylthio)chloroacetylenes react with 2-aminoethane-1-thiol, forming 2{[2-(alkylthio)ethynyl]thio}ethane-1-ammonium chlorides [124]; reactions with thiourea at room temperature give chlorides of *S*-(alkylthioethynyl) isothiuronium and *N*-[1-(alkylthio)ethylidene]thioureas [125,126]. Under comparable conditions, the reaction of (phenylthio)chloroacetylene with thiourea proceeds with the quantitative formation of the hydrochloride 4-phenylthio-1,3-thiazole-2(3*H*)–imine.

In order to obtain new heterocyclic systems with potential photochromic properties and biological activity, we were the first to study the reaction of organylthiochloroacetylenes with dialkyl-substituted thiosemicarbazones, and proved the structure of the resulting products [127,128]. Considering the bidentity of thiosemicarbazones as nucleophiles and the presence of three reaction centers in the haloacetylenes molecule-the halogen atom, terminal (C$_\alpha$) and internal (C$_\beta$) carbon atoms, the result of these reactions were not predictable.

As such, alkylthiochloroacetylenes react with thiosemicarbazones at a temperature of 20–22 °C in the medium of eponymous aliphatic ketone, forming hydrochlorides of 2-alkanone-*N*-[4-(organylthio)-1,3-thiazol-2-yl]hydrazones (**32**–**34**) (Scheme 20):

**Scheme 20.** Formation of the salts of 2-alkanone-*N*-[4-(organylthio)-1,3-thiazol-2-yl] hydrazones.

In the proton NMR spectra of **32–34**, the presence of a broad signal in the region of 9–10 ppm (Table 7), the integrated intensity of which corresponds to two protons, may indicate the protonation at the nitrogen atom. Thus, the binding of HCl in these compounds occurs at one of the nitrogen atoms (Figures 5 and 6).

**Table 7.** $^1$H and $^{13}$C NMR chemical shifts of 2-alkanone-*N*-[4-(organylthio)-1,3-thiazol-2-yl] hydrazones (**32–35**) (DMSO-$d_6$).

| | Compound | $^1$H | $^{13}$C |
|---|---|---|---|
| **32** |  | 1.20 t (3H, Me<br>1.92 s (3H, Me)<br>2.29 s (3H, Me)<br>2.88 q (2H, CH$_2$S)<br>7.14 s (1H, H-5)<br>9.93 br (2H, NH) | 13.74 (S-Me)<br>20.46 (Me)<br>24.97 (Me)<br>27.38 (SCH2)<br>106.51 C-5<br>130.12 C-4<br>161.10 C-2<br>189.43 C=N |
| **33** |  | 0.93 t (3H, Me)<br>1.18 t (3H, Me<br>1.55 m (2H, CH$_2$)<br>1.89 s (3H, Me<br>2.64 q (2H, CH$_2$)<br>2.84 q (2H, SCH$_2$)<br>7.14 s (1H, H-5)<br>9.95 br (2H, NH) | 13.74<br>20.46 (Me)<br>27.38 (Et)<br>24.97 (Me)<br>107.05 C-5<br>130.23 C-4<br>161.03 C-2<br>192.42 C=N |
| **34** |  | 1.21 t (3H, Me)<br>1.51-1.59 m<br>2.11 t, 2.50-2.60<br>cycle<br>2.81 q (SCH$_2$)<br>7.05 s (1H, H-5)<br>9.85 br (2 NH) | 13.70, 27.39 (Et)<br>24.33, 26.33, 27.39, 30.23,<br>34.81(cycle)<br>106.5 C-5<br>130.3 C-4<br>161.27 C-2<br>193.1 C=N |
| **35** |  | 1.21 t (Me)<br>1.92 s (3H, Me)<br>2.31 s (3H, Me)<br>2.87 q (SCH2)<br>7.11 s (1H, H-5)<br>9.88 br (NH) | 20.38, 24.36 (Me)<br>13.73, 26.45 Et<br>96.0 C-5<br>128.7 C-4<br>156.25 C-2<br>179.2 C=N |

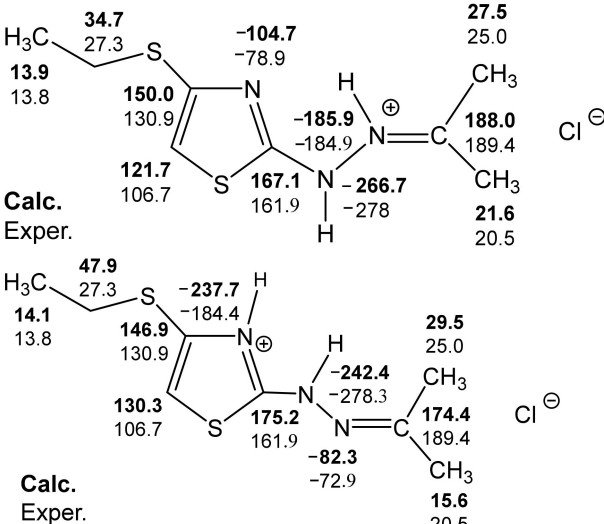

**Figure 5.** The structure of 2-propanone-*N*-[4-(ethylthio)-1,3-thiazol-2-yl]hydrazone hydrochloride (**32**).

**Figure 6.** Calculated B3LYP/6-311++G(d,p) (GIAO) and experimental $^{13}$C and $^{15}$N NMR spectra of compound **32**.

The most complete information on the structure of salts **32–34** was obtained on the basis of an analysis of the $^{13}$C, $^{13}$C(H), $^{13}$C(C) and $^{15}$N NMR spectra (Table 7). Thus, the signal in a low field at 190 ppm in the carbon spectra of these compounds is assigned to the imine carbon atom; it is split at the protons of alkyl groups with a constant $^{2}J_{CH}$ = 5.9 Hz. In addition, the analysis of satellite signals reveals two constants of spin–spin interaction (SSI) $^{13}$C-$^{13}$C, indicating the binding of this carbon atom with two magnetically nonequivalent alkyl groups. The signal in the region of 160 ppm referred to carbon C-2 (direct $^{13}$C-$^{13}$C constants are absent). The resonance signal of carbon in a higher field (106 ppm) refers to C-5–in the $^{13}$C(H) NMR spectra it splits into a doublet with a constant $^{1}J_{CH}$ = ~ 200 Hz, and has the direct $^{13}$C-$^{13}$C constant ($^{1}J_{CC}$ = 77 Hz) with the neighboring C-4 atom (Table 7) [127,128].

The $^{15}$N NMR spectrum of compound **32** contains three nitrogen signals (−72.6, −184.9, and −278.8 ppm) which refer to the nitrogen atoms of the thiazole ring, imine fragment and amino groups, respectively (Figure 5, Table 7) [44,129].

The analysis of the calculated B3LYP/6-311++G(d,p) (GIAO) and experimental $^{13}$C and $^{15}$N NMR spectra make it possible, with a sufficient degree of reliability, to give preference to the structure with a protonated imine nitrogen atom (Figure 6) [127].

Some discrepancy between the experimental and calculated values of the chemical shifts of carbon in the thiazole ring can be explained, apparently, in the following ways:

(1) the presence of a possible intramolecular hydrogen bond (**32a**), which is not fully taken into account by the used basis;

(2) the need to use a slightly different basis in the calculations of molecules containing sulfur.

**32a**

The example of compound **32** shows the possibility of obtaining neutral compounds by the dehydrochlorination of salts with an aqueous solution of sodium hydroxide (Scheme 21).

**Scheme 21.** Formation of 2-propanone-*N*-[4-(ethylthio)-1,3-thiazol-2-yl]hydrazone **35**.

As can be seen in Table 7, ongoing from a protonated molecule (**32**) to a neutral one (**35**), the resonance signals of the carbon nuclei of the thiazole ring and imine carbon, as expected, shift to a high field (3-10 ppm). The values of the direct $^{13}C$-$^{13}C$ constants of the imine carbon atom with neighboring carbon atoms indicate the *trans*- ($^1J_{CC}$ = 37 Hz) and *cis*- ($^1J_{CC}$ = 44 Hz) [45,130,131] orientation of the alkyl groups and the lone pair of the nitrogen atom relative to the C=N bond in the studied compounds, as shown by the example of compound **35** (Figure 7) [127].

**Figure 7.** Values of the $^{13}C$-$^{13}C$ constants of the imine carbon atom with carbon atoms of the methyl group.

For a more complete understanding of the features of the stereochemical structure of 2-propanone-*N*-[4-(ethylthio)-1,3-thiazol-2-yl]hydrazone (**35**), quantum chemical calculations were performed—B3LYP/6-311++G(d,p) [127]. The calculation results show that the most favorable conformation for **35** is the (syn) gauche-orientation of the lone pair of the imine nitrogen atom and the endocyclic sulfur atom (Figure 8).

**Figure 8.** Calculated B3LYP/6-311++G(d,p) (GIAO) and experimental $^{13}C$ and $^{15}N$ NMR spectra of compound **35**.

It should be noted that for the neutral molecule **35**, there is also a satisfactory agreement between the experimental and calculated values of the $^{13}C$ and $^{15}N$ NMR chemical shifts (Figure 8) (Table 7).

Thus, the analysis of the calculated-B3LYP/6-311++G(d,p) (GIAO) and experimental [13]C and [15]N NMR spectra of the two putative structures of compound **32** allows preference for the structure with a protonated imine nitrogen atom.

### 5.2. The Structure of 1,3,4-thiadiazoles and 1,3,4-thiadiazolines

The acetylation of the thiosemicarbazones 4-methoxybenzyldehyde, pyridine-4-aldehyde, thiophene-2-aldehyde, furan-2-aldehyde, indole-3-aldehyde, and isatin with acetic anhydride was studied in order to obtain new biologically active acetylamino thiadiazoles and thiadiazolines (Scheme 22) [44,45,132–135]. Here, 2-Acetylamino-1,3,4-thiadiazole (**36**) was used as a model compound.

**Scheme 22.** Formation of 2-acetylamino-1,3,4-thiadiazoles (**37**, **38**) and 2-acetylamino-4-acetyl- 5-heteryl-1,3,4-thiadiazolines (**39–44**).

The reaction is accompanied by ring closure (the addition of a mercapto group at the CH=N bond) with the simultaneous acetylation at the NH groups of the ring and NH$_2$ groups in the open chain to form 2-acetylamino-1,3,4-thiadiazoles (**37**, **38**) and 2-acetylamino-4-acetyl-5-heteryl-1,3,4-thiadiazolines (**39–44**).

The structure of these compounds was studied by multipulse and multinuclear NMR spectroscopy. The effect of the heteroaryl ring nature on the parameters of the NMR spectra of thiadiazoles was found. Here, 2-Acetylamino-1,3,4-thiadiazole (**36**) was included in the consideration in order to facilitate the identification of the reaction products (Table 8).

**Table 8.** Chemical shifts (δ, ppm) and coupling constant (*J*, Hz) in the NMR spectra of 2-acetylamino-1,3,4-thiadiazole (**36**), 2-acetylamino-5-aryl (heteryl) -1,3,4-thiadiazoles (**37**, **38**) and 2-acetylamino-4-acetyl-5-aryl(heteryl)-1,3,4-thiadiazolines (**39–44**) (DMSO-$d_6$) [132,133].

| Compound | $\delta\,{}^1H$ | $\delta\,{}^{13}C/{}^nJ(CH)$ | $\delta\,{}^{15}N$ |
|---|---|---|---|
| **36** | 2.19 s CH$_3$<br>9.13 s H-5<br>12.6 br NH | 22.49 q CH$_3$ ${}^1J$ = 129.0<br>148.54 d C-5 ${}^1J$ = 212.0<br>158.56 C-2 ${}^2J$ = 4.0<br>168.70 C=O ${}^2J$ = 6.4 | −19.9 N-4<br>−55.5 N-3<br>−242.3 NH |
| **37** | 2.21 s CH$_3$<br>3.83 s OCH$_3$<br>7.05 d H-3′,5′<br>${}^3J$ = 8.8<br>7.81 d H-2′,6′<br>${}^3J$ = 8.8<br>12.9 br NH | 22.40 d CH$_3$ ${}^1J$ = 129.4<br>55.73 d CH$_3$O ${}^1J$ = 144.6<br>115.13 C-3′,5′<br>${}^1J$ = 161.6, ${}^2J$ = 4.7<br>123.68 C-1′<br>${}^1J$ = 161.0, ${}^2J$ = 7.2<br>128.68 C-2′,6′ 158.04 C-2<br>161.52 C-4′ 162.08 C-5<br>168.83 C=O | |

**Table 8.** *Cont.*

| Compound | $\delta\,^1H$ | $\delta\,^{13}C/^nJ(CH)$ | $\delta\,^{15}N$ |
|---|---|---|---|
| **38** | 2.34 s $CH_3$<br>7.89 d H-2′,6′<br>$^3J = 5.8$<br>8.72 d H-3′,5′<br>$^3J = 5.8$<br>11.8 br NH | 22.32 $CH_3$<br>120.69 C-2′,6′<br>137.14 C-1′<br>150.61 C-3′,5′<br>159.48 C-5<br>159.53 C-2<br>168.81 C=O | −19.9 N-4<br>−55.5 N-3<br>−162.5 $N_{pyr}$<br>−242.3 NH |
| **39** | 2.03 s $CH_3$<br>2.17 s $CH_3$(NH)<br>3.72 s $OCH_3$<br>6.77 s H-5<br>6.88 d H-2′,6′<br>$^3J = 8.5$<br>7.17 d H-3′,5′<br>$^3J = 8.5$<br>11.7 br NH | 21.94 q $CH_3$ $^1J = 129.0$<br>22.45 q $CH_3$(NH) $^1J = 129.0$<br>55.22 q $OCH_3$ $^1J = 144.2$<br>65.68 C-5 dt<br>$^1J = 159.5$, $^3J = 4.0$<br>114.05 dd C-3′,5′<br>$^1J = 160.6$, $^3J = 4.8$<br>126.70 ddd C-2′,6′ $^1J = 158.2$, $^{2,3}J = 6.8,3.6$<br>133.56 dd C-1′<br>$^2J = 8.4$, $^2J = 7.2$<br>146.06 d C-2 $^2J = 4.1$,<br>159.12 d C-4′ $^2J = 8.8$<br>167.33 q C=O $^2J = 6.3$<br>169.39 qd C=O(NH)<br>$^2J = 6.5$, $^3J = 2.3$ | −113.5 N-3<br>−194.8 N-4<br>−243.8 NH |
| **40** | 2.04 s $CH_3$<br>2.24 s $CH_3$(NH)<br>6.85 s H-5<br>7.26 d H-2′,6′<br>$^3J = 8.7$<br>8.56 d H-3′,5′<br>$^3J = 8.7$<br>11.8 br NH | 21.67 q $CH_3$ $^1J = 129.0$<br>22.43 q $CH_3$(NH) $^1J = 129.8$<br>64.56 dt C-5 $^1J = 161.4$ $^3J = 4.4$<br>119.87 d C-2′,6′ $^1J = 163.6$<br>145.80 d C-2 $^2J = 4.8$<br>149.45 t C-1′ $^2J = 6.0$<br>150.06 dd C-3′,5′<br>$^1J = 179.8$, $^2J = 11.2$<br>167.66 q C=O $^2J = 6.4$<br>169.52 q C=O (NH) $^2J = 6.5$ | −113.6 N-3<br>−172.4 $N_{pyr}$<br>−198.8 N-4<br>−243.6 NH |
| **41** | 2.06 $CH_3$<br>2.16 $CH_3$ (NH)<br>6.94 H-4′<br>7.07 H-3′<br>7.11 H-5<br>7.44 H-3′<br>11.7 br NH | 21.71, 22.48<br>61.46 C-5<br>125.23 C-5′<br>126.11 C-4′<br>126.74 C-3′<br>144.54 C-2′<br>146.12 C-2<br>167.22, 169.43 C=O | −115.1 N-3<br>−199.2 N-4<br>−243.4 NH |

**Table 8.** *Cont.*

| Compound | $\delta\,{}^1H$ | $\delta\,{}^{13}C/{}^nJ(CH)$ | $\delta\,{}^{15}N$ |
|---|---|---|---|
| **42** | 2.08 CH$_3$<br>2.21 CH$_3$ (NH)<br>6.31 H-3′<br>6.39 H-4′<br>6.90 H-5<br>7.59 H-5′<br>11.6 br NH | 21.86 q CH$_3$ ${}^1J = 129.4$<br>22.56 q CH$_3$ (NH) ${}^1J = 129.4$<br>59.38 q C-5 ${}^1J = 159.8$<br>107.19 dt C-4′<br>${}^1J = 176.6,\ {}^2J = 3.2$<br>110.69 ddd C-3′<br>${}^1J = 176.1,\ {}^{2,3}J = 13.6,\ 4.0$<br>143.17 ddd C-5′<br>${}^1J = 204.5,\ {}^{2,3}J = 11.2,\ 7.6$<br>145.87 d C-2 ${}^2J = 4.8$<br>151.60 dt C-2′<br>${}^2J = 17.2,\ {}^2J = 7.2$<br>167.42 q C=O ${}^2J = 6.4$<br>169.55 C=O (NH) ${}^2J = 6.4$ | −117.1 N-3<br>−196.2 N-4<br>−246.4 NH |
| **43** | 2.07, 2.15 CH$_3$<br>7.00 H-6′<br>7.11 H-5′<br>7.15 H 5<br>7.30 H-2<br>7.38 H-4′<br>7.49 H-7′<br>11.09 NH-indol<br>11.7 br NH | 22.02 q CH$_3$ (N-4) ${}^1J = 129.0$<br>22.69 q CH$_3$ (NH) ${}^1J = 129.4$<br>60.99 d C-5 ${}^1J = 158.2$<br>111.93 dd C-7′ ${}^1J = 159.4,\ {}^2J = 7.6$<br>114.94 C-3′ ${}^2J = 8.4,\ {}^2J = 6.0$<br>118.80 dd C-5′ ${}^1J = 158.2,\ {}^2J = 7.6$<br>119.15 dd C-6′ ${}^1J = 158.6,\ {}^2J = 7.2$<br>121.49 dd C-4′ ${}^1J = 158.2,\ {}^2J = 7.6$<br>123.63 dd C-2′ ${}^1J = 182.6,\ {}^2J = 5.2$<br>123.99 m C-9′<br>136.82 dd C-8′ ${}^2J = 9.2,\ {}^2J = 3.2$<br>146.76 C-2 ${}^2J = 4.9$<br>167.23 q C=O ${}^2J = 6.4$<br>169.32 q C=O (NH) ${}^2J = 6.8$ | −116.6 N-3<br>−192.3 N-4<br>−243.2 NH<br>−245.4 NH |
| **44** | 2.10 CH$_3$(N-4)<br>2.15 CH$_3$(NH)<br>2.56 CH$_3$<br>7.27 dd H-6′<br>${}^3J = 8.1,\ {}^3J = 7.3$<br>7.41 dd H-5′<br>${}^3J = 7.3,\ {}^3J = 7.2$<br>7.47 d H-4′ ${}^3J = 7.2$<br>8.08 d H-7′ ${}^3J = 8.2$<br>12.0 br NH | 21.86 q CH$_3$ (N-4) ${}^1J = 129.8$<br>22.33 q CH$_3$ (NH) ${}^1J = 129.0$<br>26.05 q CH$_3$ (N-1′) ${}^1J = 130.6$<br>75.13 d C-5 ${}^3J = 3.2$<br>115.72 dd C-7′<br>${}^1J = 169.4,\ {}^2J = 7.2$<br>123.96 dd C-6′<br>${}^1J = 164.6,\ {}^2J = 8.8$<br>125.88 dd C-5′ | −115.9 N-3<br>−192.4 N-4<br>−200.6 N-1′<br>−245.5 NH |

In the ${}^1$H NMR spectrum of the model 2-acetylamino-1,3,4-thiadiazole (**36**) (Table 8), the downfield signal (9.13 ppm) refers to the proton at position 5, while the ${}^{13}$C(H) spectrum shows a significant splitting signal C-5, characteristic of such cycles (${}^1J_{CH}$= 212 Hz). As can be seen from the table, the signal of the protons of the methyl group of the NHCOCH$_3$-fragment in the studied compounds is in a more downfield field (2.1–2.3 ppm) than the protons of the acetyl group (2.03–2.10 ppm), and does not almost depend on the substituent in position 5 of the thiadiazole moiety. The values of the chemical shifts of the NH-group proton of all of the studied compounds are also practically insensitive to the effect of the hetero ring substituent (11.6–12.9 ppm) [132,133].

The resonance position of both H-5 and C-5 thiadiazolines **39**–**44** weakly depends on the nature of the substituent in this position ($\Delta\delta$ 4 and 6 ppm, respectively), while in thiadiazole **36**–**38** (more aromatic system), the effect of the substituent turns out to be significant: when both pyridine (**38**) and aryl substituents (**37**) are introduced into position 5, the values of $\Delta\delta$ (C-5) are 12 ppm.

Thus, in thiadiazole derivatives, the chemical shifts of the hetero ring nuclei are more sensitive to the effect of the substituent than those in thiadiazolines.

## 6. Conclusions

The vicarious nucleophilic substitution of hydrogen is the key step in the formation of purine systems from commercial nitroimidazoles. There is no doubt that the VNS of hydrogen will occupy a rightful place in the synthetic arsenal of researchers and will provide a better insight into the fine chemical processes occurring in biologically active compounds. The vicarious amination of nitroazoles is facile, and is almost the only approach to introduce an amino group into the azole ring. The theoretical study of the stereochemical behavior of azoles in combination with experimental methods of multinuclear and multipulse NMR spectroscopy is indispensable for the understanding of the subtle structural aspects of the compounds that allow their biological activity to be predicted. In recent years, interest in fuctionalized azoles and related nitrogen-rich compounds has steadily increased, as evidenced by the large number of publications in the field of their chemical and structural studies [136–141]. Nitrogen NMR spectroscopy is an appropriate, powerful, reliable and convenient technique for the determination of the stereochemical features of nitrogen-containing heteroaromatic compounds.

**Funding:** This research received no external funding.

**Data Availability Statement:** The study did not report any data.

**Acknowledgments:** I thank Igor Grushin for his support and kind assistance in translating this manuscript. I am very grateful to my extended family for their endless support all these years, when we were left alone so early without our main family member, who passed away in an untimely manner; Larin Mikhail Fedorovich was a well-known specialist in the field of NMR spectroscopy of organoelement compounds. And so, my big and friendly family: daughter Polina, son Sergei, parents Polina and Ivan, sister Tatyana, brother Alexei, daughter-in-law Maria, son-in-law Boris, as well as all the grandchildren Mikhail, Anastasia, Anna, Stepan, Alexander, Leonid, Anatoly, Alena and Arina.

**Conflicts of Interest:** The authors declare no conflict of interest.

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
