# Peer review of "The Structure of Biologically Active Functionalized Azoles: NMR Spectroscopy and Quantum Chemistry"

_magnetochemistry, doi:10.3390/magnetochemistry8050052_

Round 1

Reviewer 1 Report

The manuscript by Lyudmila I. Larina represents a comprehensive review detailing an important topic of NMR spectroscopic assignment in functionalized azoles.  The article rigorously summarizes a broad array of 1H, 13C and 15N NMR spectroscopic results and attempts to rationalize them along with the outcome of quantum chemical research. The review offers very thorough coverage of existing literature and to the moment is likely the most complete compendium on the subject. In a view of sharply growing interest in the development of effective methods of identification and characterization of functionalized azoles as pharmaceuticals and advanced materials in general, the topic of the manuscript is of considerable interest to a rather broad audience, including synthetic organic chemists and material scientists. The manuscript is well structured, the language is clear and concise, and the material is reported in a logical and easy-to-follow manner. The length of the manuscript and the number of tables and illustrations are completely adequate.

In a conclusion, this is a very informative and well-written review paper devoted to a prospective approach to spectroscopic identification in functionalized azoles and can be positively recommended for publication in Magnetochemistry.

Author Response

Author response 1
Thank you very much for your comprehensive analysis of my review and for kind appreciation of my
work. My best wishes to the Reviewer 1.

Reviewer 2 Report

The review article of L. I. Larina presents research on chemistry of azoles, focusing mainly on NMR techniques supported by quantum computations. Due to the high relevance of azoles in a variety of disciplines, the presently discussed topics merit a review article, even more so because of the high capability of magnetic techniques to address many structural issues crucial for the understanding of the functionality of azoles. In that sense, publication of the present article is encouraged.

Of the very few critical points that I could raise, I’d like to stress the following: the relatively low energy characteristic of transitions between nuclear spin states (observed in the RF domain) implies that the NMR experiment probes the matter at relatively long timescales, effectively averaging much of conformational isomerism or even chemical modifications (e.g., tautomerism) characterized by shorter timescales, i.e. with low-enough barriers. When multiple conformers/isomers are considered by quantum calculations, caution is needed when several isomers are found to have comparable energies and are separated by low barriers – the observable chemical shifts should possibly be estimated as Boltzmann-weighted average of the corresponding individual structures. Has this “principle” been taken into account in the presented research involving quantum chemistry?

In addition, the way in which vicarious nucleophilic substitution is presented (beginning of Section 2, pp. 3-4) may appear slightly confuzing to an unfamiliar reader. Perhaps it should be first summarized how VNC is related to azole chemistry before explaining it in a great detail. Also, the acronym VNC should be given where the corresponding term firstly appears in the text and use it consistently from that point on.   

Author Response

Reply to Reviewer 2
The review article of L.I. Larina presents research on chemistry of azoles, focusing
mainly on NMR techniques supported by quantum computations. Due to the high
relevance of azoles in a variety of disciplines, the presently discussed topics merit
a review article, even more so because of the high capability of magnetic
techniques to address many structural issues crucial for the understanding of the
functionality of azoles. In that sense, publication of the present article is
encouraged.
Of the very few critical points that I could raise, I’d like to stress the following: the
relatively low energy characteristic of transitions between nuclear spin states
(observed in the RF domain) implies that the NMR experiment probes the matter at
relatively long timescales, effectively averaging much of conformational
isomerism or even chemical modifications (e.g., tautomerism) characterized by
shorter timescales, i.e. with low-enough barriers. When multiple
conformers/isomers are considered by quantum calculations, caution is needed
when several isomers are found to have comparable energies and are separated by
low barriers – the observable chemical shifts should possibly be estimated as
Boltzmann-weighted average of the corresponding individual structures. Has this
“principle” been taken into account in the presented research involving quantum
chemistry?
Author response 1
Thank you very much for a very important question. Yes, of course, the Boltzmann
principle is taken into account. At the same time, a high-level theory is applied to
calculate the energy of tautomers; we calculated using the MP2 theory.
In addition, the way in which vicarious nucleophilic substitution is presented
(beginning of Section 2, pp. 3-4) may appear slightly confuzing to an unfamiliar
reader. Perhaps it should be first summarized how VNC is related to azole
chemistry before explaining it in a great detail.
Author response 2
Thank you very much. Thank you very much. I have followed the defined
approach. At the beginning of the first chapter, I made a brief introduction to the
principle of aromatic nucleophilic substitution and then moved on to the concept of
vicarious nucleophilic substitution of hydrogen, and in particular vicarious Camination.
Then I wrote that vicarious C-amination is practically the only way to
introduce an amino group into nitroazoles. Functionalized nitroaminoazoles are
high-energy compounds that are so necessary both in the energy sector and in the
drug industry.
Also, the acronym VNC should be given where the corresponding term firstly
appears in the text and use it consistently from that point on.
Author response 3
Yes, thank you, it really needs to be done first. I have corrected this.
